# Review on the Lymphatic Vessels in the Dental Pulp

**DOI:** 10.3390/biology10121257

**Published:** 2021-12-02

**Authors:** Kamila Wiśniewska, Zbigniew Rybak, Maria Szymonowicz, Piotr Kuropka, Maciej Dobrzyński

**Affiliations:** 1Department of Dental Surgery, Wroclaw Medical University, Krakowska 26, 50-425 Wroclaw, Poland; 2Pre-Clinical Research Centre, Wroclaw Medical University, Bujwida 44, 50-345 Wroclaw, Poland; zbigniew.rybak@umw.edu.pl (Z.R.); maria.szymonowicz@umw.edu.pl (M.S.); 3Department of Histology and Embryology, Wroclaw University of Environmental and Life Sciences, Norwida 25, 50-375 Wroclaw, Poland; piotr.kuropka@upwr.edu.pl; 4Department of Pediatric Dentistry and Preclinical Dentistry, Wroclaw Medical University, Krakowska 26, 50-425 Wroclaw, Poland; maciej.dobrzynski@umw.edu.pl

**Keywords:** histology, human dental pulp, immuhistochemistry, lymphatic and blood markers, lymphatic vessels

## Abstract

**Simple Summary:**

It is debatable whether lymphatic vessels exist in the dental pulp. Most researchers confirm their presence; however, the lymphatic system in the dental pulp is much less developed compared to other tissues of the body. Lymphangiogenesis occurs in the dental pulp with inflammatory changes as a response to inflammatory stimuli acting on the tooth. If lymphangiogenesis is defined as the development of lymphatic vessels from already existing ones, such a mechanism is possible only when lymphatic vessels are present in healthy teeth. Research papers have not conclusively proved whether lymphatic vessels can form in the dental pulp. The use of an immunohistochemical examination can very likely prove the presence of a lymphatic system in dental tissues. However, the evaluation of the lymphatic system of the teeth is problematic because it is quite difficult to clearly distinguish lymphatic vessels from small blood vessels.

**Abstract:**

Despite many studies, opinions on the lymphatic system of the teeth are still incompatible. Studies using light and electron microscopy and directly using methods such as a radioisotope (radionuclide) scan and interstitial fluid pressure measurement reported incomplete results. Immunohistochemistry (IHC) plays the main role in investigating presence of the lymphatic system in dental tissues. This method uses labeled antibodies against antigens typical of lymphatic vessels. The use of appropriate staining enables the detection of antigen-antibody reaction products using a light (optical), electron or fluorescence microscope. However, these studies do not show the system of vessels, their histologic structure under physiological conditions and inflammation as well as the lymphangiogenesis process in the dental pulp. Unfortunately, there is a lack of studies associating the presence of lymphatic vessels in the dental pulp with local lymphatic nodes or large vessels outside the tooth. In the scientific and research environment, the evaluation of the lymphatic system of the teeth is problematic because it is quite difficult to clearly distinguish lymphatic vessels from small blood vessels. Despite many indications of the presence of lymphatic vessels in the pulp chamber, this problem remains open and needs further research.

## 1. Lymphatic System in the Body

The lymphatic system is an open circulatory system for lymph fluid. The lymphatic system consists of lymphatic vessels, lymph nodes and accessory organs, including the bone marrow, spleen, thymus and tonsils. Lymph is formed in the tissues from intercellular fluid and then flows towards the capillaries, larger lymphatic trunks and lymph nodes, reaching the thoracic duct that ascends to the left subclavian vein [1,2,3,4]. Lymph mediates the exchange of components between blood and other body tissues, transports fats and helps remove excess fluids and toxins from the body. Lymphocytes are formed and then they mature in the lymphatic system. Lymphatic vessels remove cellular debris and bacteria from the site of inflammation. Excess fluid and large macromolecules, such as plasma proteins, are removed from the tissues and return to the blood [5,6]. The presence of valves that prevent the backflow of lymph is a significant feature of lymphatic vessels [7]. The main function of lymphatic vessels is the reabsorption of fluids from the interstitial space back into the circulatory system. This involves the transport of proteins, including antibodies and cells [1].

Research concerning the lymphatic system began in the 17th century. Light microscopy revealed thin white vessels distributed in the mesentery and over the upper intestinal wall. The vessels were called “venae albae aut lacteae”: “venae” due to their resemblance to veins, “albae” to distinguish them from blood vessels, and “lacteae” due to the milk-like fluid they contain [8]. Lymphatic valves were visualised; the thoracic duct and its opening into the left subclavian vein were found. Lymph flow into the circulatory system was also described. Lymph nodes were found to act as filters in the lymphatic system. Lymphatic fluid from any area of the body drains through the lymphatic system to a specific lymph node and then to other lymph nodes. Those studies developed a better understanding of lymphatic anatomy and lymphatic circulation [9,10].

Lymphatic vessels are found in most tissues of the body. The head and neck region has an extensive network of lymphatic vessels and one-third of all human lymph nodes. Lymph containing antigens from the periodontium and teeth flows to the submandibular—anterior, medial, and posterior—lymph nodes and to the submental lymph nodes. The lymph is then transported to the deep jugular nodes, whose efferent vessels form the jugular trunk from where it enters the venous angle formed by the junction of the internal jugular vein and the subclavian vein [2].

The fibres that are present in soft tissues and anchor the initial lymphatic vessels are composed of fibrillin and resemble the elastic microfibres with which they are connected. They enable tension to be transferred to the surrounding collagen fibres, ensuring elongation and relaxation of lymphatic vessels. This phenomenon is associated with lymph flow [4,11]. The presence and proper function of lymphatic vessels appears to depend on the distribution of local connective tissue.

## 2. Visualisation of Lymphatic Vessels in Soft Tissues

Most lymphatic vessels are located in soft tissues of the human body; hence, there are many methods that have been applied to their evaluation. The current standard for the imaging of lymphatic flow is the use of isotopes in lymphoscintigraphy. Indicators based on radionuclides are most frequently used. This leads to low-resolution images and the vessels are not clearly visible [7,12,13]. None of the known lymphoscintigraphy methods have undergone the standardisation required for daily use, as many hospitals use different radioisotopes with varying radioactivity under various data interpretation standards [14].

Fluorescence lymphography is an alternative method for determining lymph flow in lymphatic vessels. Indocyanine green is water-soluble and has been widely used in the evaluation of cardiac output, liver function and angiography for more than half a century. It contains sodium iodide; thus, it should be used with caution in patients with a history of iodine allergy [7,14]. Fluorescence lymphography is a safe and economical method that is less invasive than lymphoscintigraphy. It enables measurements whether sitting or standing. However, only vessels at the skin surface are detected, making it difficult to visualise deep vessels that run in interstitial spaces. Compared to traditional lymphoscintigraphy, the equipment used in fluorescence lymphography is less expensive, minimally invasive and provides real-time imaging. Unno et al. [14] measured the dye flow time in the lower limb during the standing position, supine position, standing position during muscle massage, and standing position while performing exercise on an ergometer exercise bike (50 W at 50 rpm./min.) in healthy volunteers for 14 days. The study revealed that the lymph flow time was shorter in the supine position compared to the standing position. The standing position creates hydrostatic pressure that causes lymph to flow against gravity. The lymph flow time was shorter during muscle massage and during exercise because the lymph flow was forced.

Laser Doppler flowmetry (LDF) is an effective tool for monitoring pulp blood flow and makes it possible to check the pulp vitality [15]. Suzuki et al. [12] performed an ultrasound examination of blood flow in the blood vessels in parallel with the lymphatic system tests. Moreover, they performed plethysmography, i.e., the measurement of changes in blood flow through peripheral vessels close to the body surface area. The aforementioned examinations proved that lymphatic vessels have a parallel course of blood vessels. The difficulty in assessing the presence of lymphatic vessels in the dental pulp is caused by the hard tissue environment of the tooth.

## 3. Characteristics of the Dental Pulp

The dental pulp is a specific type of connective tissue. It contains cells, fibres, intercellular substance, nerve endings, blood and lymph vessels. There are numerous veno-venous and arterio-venous anastomoses between the vessels in the pulp, leaving the area open and independent of a single source of blood supply. They provide nutrition to the tissues and equalise pressure in the excretory vessels during inflammatory processes [16].

The dental pulp is surrounded by rigid walls of dentine, a layer of enamel and root cementum. Some specific situations require increased vessel density. However, excessive blood vessel growth can be harmful and lead to pulpitis. This is due to the limited space of the pulp chamber of the tooth. When the fluid pressure in the dental pulp increases until it reaches vascular pressure, this can result in ischaemia and risk of necrosis of the dental pulp [17,18].

Unlike lymphatic vessels in other tissues, lymphatic vessels in the dental pulp are not surrounded by elastic fibres from the surrounding connective tissue; only a few anchoring fibres are present. Therefore, lymphatic vessels are directly connected to surrounding collagen fibres, forcing them to resist intracellular pressure and remain open [19,20]. A drop in blood pressure in the pulp is relatively small [21] compared to, for example, skeletal muscles [22]. Changes in circulation in adjacent tissues, such as the periodontal ligament, gingiva and alveolar bone, can affect blood circulation in the dental pulp.

Studies have found that tooth movement during chewing acts as an additional external stimulator of lymph flow in lymphatic vessels, leaving the pulp and reaching the periodontal ligaments. Lymphatic vessels found in the periodontal ligament area run towards the alveolar bone and between the teeth, connecting with one another [11,23].

During tooth development, the dentine-forming cells—odontoblasts—of the pulp-dentine complex produce primary dentine, which is the main material of the tooth. In a mature tooth, the process of dentine formation continues. As the body ages, the dental pulp accumulates more layers of dentine. It is initially responsible for the formation of the primary pulp. In a fully formed tooth, it forms the secondary dentine. Aging reduces the dimensions of the pulp chamber due to the formation of tertiary dentine and atrophy of the dental pulp. It begins with apoptosis of odontoblasts, fibroblasts, and the accumulation of hydroxyapatite crystals, and it is accompanied by decreased perfusion due to the narrowing of the apical foramen by dental cementum deposits. At the same time, the number and diameter of nerve fibres decreases. Blood vessels also become calcified [16].

The tertiary dental pulp is also formed as a defensive reaction of the dental pulp to external irritants such as tooth surface abrasion, caries development and preparations for fillings. Ultimately, this leads to the thickening of the mineralised barrier that separates the pulp chamber from the oral microbial environment [24,25].

In teeth with caries, dilated vessels with a structure typical of lymphatic vessels were found, i.e., thin walls, irregular shape, no erythrocytes in the lumen of the lymphatic vessel [26]. Dental pulpitis additionally causes deposition of damaged secondary dentine or deposition of mineral salts in the form of free denticles (pulp stones) or pulp stones adhering to hard tissues (adherent denticles) [27]. The more mature the teeth, the smaller the pulp chamber. This is related to the deposition of secondary dentine. Hence, the thesis that the lymphatic system of the teeth can develop with aging when the dimensions of the surrounding tissues continue to decrease must be considered questionable [28].

Deep carious lesions are the most common cause of dental pulpitis. However, there are indications that carious bacteria may enter the dental pulp from the blood vessels as a result of septicaemic phenomena [29]. The inflammation is accompanied by increased angiogenesis; lymphatic vessels may also appear [30]. To date, it has not been proved whether the lymphatic vessels in the pulp are in contact with the adjacent lymphatic vessels or they are a secondary response to increased angiogenesis due to the inflammatory process. Lymphatic vessels are most frequently formed from already existing ones. The formation of lymphatic vessels in response to inflammation is a matter of dispute [31].

### 3.1. Development of the Inflammatory Process in the Dental Pulp

When the enamel and dentine surfaces are damaged by a deep carious lesion or a dental tissue fracture due to trauma, bacterial colonisation of the root canals can occur, as well as formation of biofilm that infects the dental pulp [32,33,34]. Bacteria are very frequently subject to colonisation in spaces beyond the reach of dental instruments and antimicrobials. Proteins from remaining necrotic tissues and bacterial adhesives provide nutrients for their survival [35]. In severe caries, dentinal tubules are the main route of discharge of bacterial metabolites such as lactic acid, propionic acid and acetic acid [36]. At low pH due to the presence of the aforementioned acids, a specific byproduct is activated—lipoteicholic acid (LTA). LTA is extracellularly released by acidogenic Gram-positive bacteria [36,37].

Odontoblasts show expression of toll-like receptors that recognise specific pathogens and trigger the recruitment of other immune cells, such as dendritic cells and lymphocytes [38,39]. Other molecules, including β-defensin [40] or transforming growth factor β (TGF-β), are also released by odontoblasts. β-defensin has a bactericidal effect on bacteria that are commonly involved in caries, such as *Streptococcus mutans* and *Lactobacillu ssp.* [40]. TGF-β controls homeostasis by exerting pro-inflammatory effects in the early stage of injury, and anti-inflammatory and profibrotic effects in the late stage of injury [41,42].

Reuptake of extravasated fluids in the dental organ is of great importance in terms of the process of maintaining homeostasis of the dental pulp and the nerves and odontoblasts contained in it. The lymphatic system provides an alternative means of recirculating intercellular fluid while becoming a defensive element against potential bacterial infection associated with pathological processes in the oral cavity. The understanding of the mechanisms of angiogenesis and lymphangiogenesis during these processes may be beneficial for more effective disease management; however, further research is necessary.

Currently, the standard for identification of lymphatic vessels and imaging of lymph flow is the use of isotopes in lymphoscintigraphy. In the case of the dental pulp, however, the usefulness of this method is limited due to the specificity of the structure and location of dental pulp.

Inflammation-induced lymphangiogenesis is a widely accepted concept; however, the intrinsic mechanisms concerning the interaction between inflammatory cells and lymphatic endothelial cells are not fully understood [31]. Macrophages, monocytes and neutrophils were found to be involved in inflammation-induced lymphangiogenesis, but the role of mast cells, plasma cells and eosinophils is still a matter of dispute [31,43,44,45].

In lymphangiogenesis, after exposure to inflammatory stimuli, the best described signaling pathways include: vascular endothelial growth factor C/vascular endothelial growth factor receptor-3 (VEGF-C/VEGFR-3) and vascular endothelial growth factor A/vascular endothelial growth factor receptor-2 (VEGF-A/VEGFR-2) [46,47,48,49,50,51].

The healing process of the dental pulp is associated with the expression of several proangiogenic growth factors, such as platelet-derived growth factor (PDGF), vascular endothelial growth factor (VEGF), fibroblast growth factor 2 (FGF-2), angiogenin, angiopoietin, epidermal growth factor, heparin-binding epidermal growth factor, hepatocyte growth factor, leptin, and placental growth factor. Many of these proangiogenic growth factors may also directly or indirectly stimulate lymphangiogenesis [52,53].

### 3.2. The Presence of Lymphatic Vessels in the Dental Pulp of Various Animal Species

Despite many studies, the dental lymphatic system is still debated due to the new methods implemented in the research. The first studies were based on light and electron microscope, radioisotope scanning and tissue fluid pressure measurement [23].

Some of the investigations proved the existence of lymphatic vessels in the dental pulp based on the characteristic lumen structure of lymphatic vessels by means of transmission electron microscopy [18], characteristics of lymphatic vessels in inflammatory dental pulp [20], enzyme-histochemical staining using 5′nucleotidase [54] or immunohistochemical staining [55]. Some of the reports denied the presence of lymphatic vessels in the dental pulp [28]. Martin et al. (2010) suggested that capillary blood vessels and the tissue interstitium may serve as a closed circulatory system such as lymphatic vessels [28]. Gerli et al. (2010) reported that the presence of lymphatic vessels is probably a false-positive caused by fiber bundles and inflammatory cells which were stained due to the lack of blocking [56].

In recent years, immunohistochemical markers, which are characteristic of lymphatic vessel endothelium, have become available on the market [28,57,58]. Their application along with the use of electron microscope and enzyme-histochemical procedures allow to confirm the presence of lymphatic vessels [19,59].

Table 1, Table 2, Table 3, Table 4 and Table 5 present the studies of lymphatic vessels in the structures of the tooth in various species with different evaluation techniques.

## 4. Molecular Markers of Lymphangiogenesis

Lymphangiogenesis is regulated by vascular endothelial growth factors: VEGF-C and VEGF-D, and their receptors: VEGFR-2, VEGFR-3, neuropilin-2 (VEGFR-3 co-receptor), angiopoietic factors and their receptors. Endothelial cells of lymphatic vessels are characterised by the expression of the Prox-1 transcription factor, the presence of molecules such as Lymphatic Vessel Endothelial Receptor 1 (LYVE-1), podoplanin (PDPN) and integrin α2 (adhesion molecule). The aforementioned lymphatic endothelial cell markers can also be found in other cells. Therefore, these markers are not exclusive to lymphatic vessels. The immunohistochemical evaluation of lymphangiogenesis uses preferably a combination of lymphatic markers (Anti-Prox 1, Anti-VEGFR-3, Anti-LYVE1, Anti-PDPN). They enable an objective evaluation of lymphangiogenesis. Transcription factors, such as Prox-1, regulating the differentiation of precursor lymphatic vessels (Table 6). Lymphatic endothelial cell differentiation is also influenced by, i.a., VEGF-C, VEGF-D, angiopoietin-2, transmembrane ligand—ephrin-B2, neuropilin-2 receptor protein and transmembrane glycoprotein—podoplanin. At the inflammatory site, lymphangiogenesis is regulated by the same factors that affect lymphatic vessel development in terms of individual development [57,93].

The hyaluronic acid receptor LYVE-1 is expressed by the endothelium of lymphatic vessels on the lumen and tissue sides. LYVE-1 promotes receptor-dependent endocytosis of hyaluronic acid, suggesting the involvement of LYVE-1 in transcytosis of macromolecules [4,54,94,95]. Studies show that growth factors such as PDGF, VEGF-A, VEGF-C, VEGF-D and human placental growth factor (human PGF), containing the cell-surface retention sequence (CSR) that can be found in hyaluronic acid, are LYVE-1 ligands (antibodies directed against the first endothelial receptor for hyaluronic acid). Following the formation of the aforementioned LYVE-1 ligand complex, the cell-surface retention sequence binding protein-1 (CRSBP-1) is phosphorylated, initiating a signaling cascade that results in the opening of lymphatic endothelial intercellular junctions [94]. When choosing LYVE-1 as a marker in the evaluation of lymphangiogenesis, it should be noted that the LYVE-1 molecule can be degraded during the inflammatory process [96].

Prospero-related homeobox 1 (Prox-1) is a transcription factor located in the nucleus of lymphatic endothelial cells [97,98]. The released Prox-1 factor determines the differentiation of lymphatic endothelial cells, the maintenance of the characteristics of mature lymphatic vessels, as well as the proliferation and differentiation of other cell types (e.g., nerve cells). It is also involved in the induction of the expression of E1 cytokines, which regulates cell migration from G1 phase to S phase. Anti-Prox-1 antibodies selectively detect lymphatic endothelial cells in various animal species against the transcription factor [11,98,99]. During Prox1 labeling, Martin et al. [28]. did not find any positive nuclear immune response in 2145 pulp preparations in three sections: apical, medial and occlusal areas of canine teeth.

The VEGFR-3, also known as Flt4 (fms-related tyrosine kinase 4), is selectively present on lymphatic endothelial cells; it is also expressed on the endothelial surface of blood vessels. The Flt4 was first detected on the surface of lymphatic vessels [100,101,102]. It plays a central role in the regulation of lymphangiogenesis, as it is a signaling receptor for VEGF-C and VEGF-D—growth factors that are specific to lymphatic vessels. The receptor stimulation activates the proliferation and migration of lymphatic endothelial cells. It also prevents apoptosis [103,104,105]. In mature venous vessels, VEGFR-3 disappears on the surface of endothelial cells and becomes specific to lymphatic vessels. The VEGF gene expression by pulp fibroblasts is induced by bacteria such as *Pseudomonas endodontalis*, *Pseudomonas gingivalis* and *Pseudomonas intermedia*, which may be associated with the development of inflammation within the dental pulp by stimulating VEGF products [106].

Infected tissues enhance the expression of inflammatory mediators. It has been noted that this expression is strongly influenced by cytokines that induce the VEGF mRNA gene expression in human dental pulp and human gingival fibroblasts. This may partially contribute to the damage of dental pulp and periapical tissues by expanding the vascular network, which in turn could increase inflammation. Lipopolysaccharides (LPS) produced by Gram-negative (G−) bacteria and lipoteichoic acid (LTA) produced by Gram-positive (G+) bacteria under VEGF expression pattern also contribute to inflammation. These bacteria are found in deep carious lesions and in reversible pulpitis. Lipopolysaccharides determine cytotoxicity, pyrogenicity and macrophage activation [107].

Podoplanin is a transmembrane protein that appears on the surface of lymphatic endothelial cells on the lumen side and in small amounts in the tissue area. It is also found on the surface of other cells, e.g., osteocytes. Podoplanin enables lymphatic vessels to mature. It is recognised by the marker D2-40, which enables its identification by immunohistochemical examinations [59]. Angiopoietin and its receptor contribute to lymphangiogenesis; however, their function is not fully understood [108]. Despite many indications of the presence of lymphatic vessels in the dental pulp, this problem is still open and requires further research.

## 5. Conclusions

The presence of lymphatic vessels in the dental pulp is a matter of dispute. Most researchers confirm their presence; however, the lymphatic system in the dental pulp is much less developed compared to other tissues of the body.

Lymphangiogenesis occurs in the dental pulp with inflammatory changes as a response to inflammatory stimuli acting on the tooth. If lymphangiogenesis is defined as the development of lymphatic vessels from already existing ones, such a mechanism is possible only when lymphatic vessels are present in healthy teeth. Research papers have not conclusively proved whether lymphatic vessels can form in the dental pulp. The use of an immunohistochemical examination is very likely to prove the presence of lymphatic system in dental tissues. However, the evaluation of the lymphatic system of the teeth is problematic because it is quite difficult to clearly distinguish lymphatic vessels from small blood vessels. Understanding the mechanisms of angiogenesis and lymphangiogenesis during these processes can be beneficial for more effective treatment of diseases, but further research is needed.

## Figures and Tables

**Table 1 biology-10-01257-t001:** Examination of the presence of lymph vessels in the dental pulp with the use of a light microscope [60].

Author, Year	Species	Lymphatic Vessel in the Dental Pulp
Schweitzer, 1907 [61]	rabbit, monkey, dog	+
Magnus, 1922 [62]	human, ox	+
Fish, 1927 [63]	dog	+
Noyes et al., 1929 [64]	rabbit, dog	+
MacGregor, 1936 [65]	cat, monkey, dog, guinea pig	+
Sulzmann, 1955 [66]	dog	+
Balogh et al., 1957 [67]	human	±
Isokawa, 1960 [68]	dog	−
Bernick, 1977b [69]	dog	+
Brown et al., 1969 [70]	dog	+
Ruben et al., 1971 [71]	dog	+
Dahl et al., 1973 [72]	human	−
Bernick, 1977b [69]	human (healthy pulp)	+
Bernick, 1977a [26]	human (inflamed pulp)	+
Frank et al., 1977 [73]	human	+
Rodd et al., 2003 [74]	human	+

Observation: “+” positive result, “−” negative result, “ ± “ equivocal result.

**Table 2 biology-10-01257-t002:** Examination of the presence of lymph vessels in the dental pulp with the use of a microscope electron microscope [60].

Author	Species	Lymphatic Vessel in the Dental Pulp
Kukletová, 1979 [75]	calf	+
Dahl, 1973 [72]	human	+
Gängler et al., 1980 [76]	human, rat, cat, dog	−
Vongsavan et al., 1992 [77]	cat	−
Marchetti, 1996 [78]	human	+
Qi et al., 2000 [79]	human	+
Zhang et al., 2000 [80]	human	+

Observation: “+” positive result, “−” negative result.

**Table 3 biology-10-01257-t003:** Examination of the presence of lymph vessels in the dental pulp using a light and electron microscope [60].

Author	Species	Lymphatic Vessel in the Dental Pulp
Eifinger, 1970 [81]	human	±
Takada, 1973 [82]	dog, mice, guinea pig, rabbit, human	−
Bishop, 1990 [18]	cat	+
Marchetti et al., 1990 [83]	human	+
Marchetti et al., 1991 [84]	human	+
Marchetti, 1992 [20]	human	+
Marchetti et al., 2002 [85]	human	+
Oehmke, 2003 [19]	human	+

Observation: “+” positive result, “−” negative result, “ ± “ equivocal result.

**Table 4 biology-10-01257-t004:** Examination the presence of lymphatic vessels in the dental pulp method enzyme histochemistry [60].

Author	Species	Lymphatic Vessel in the Dental Pulp
Aoyama et al., 1995 [54]	human	+
Aoyama, 1996 [86]	human	+
Matsumoto et al., 1997 [87]	human	+
Matsumoto et al., 2002 [88]	rat, hamster, monkey, human	+

Observation: “+” positive result.

**Table 5 biology-10-01257-t005:** Examination of the presence of lymphatic vessels in the dental pulp using the immunohistochemical method [60].

Author	Species	Method	Lymphatic Vessel in the Dental Pulp
Sawa et al., 1998 [89]	human	light microscope, IHC: mAb-D, anti-L	+
Pimenta et al., 2003 [90]	human	light microscope, IHC: anti-VEGFR-3, anti-CD31	+
Berggreen, 2009 [23]	mice, rats	light and fluorescence microscope, IHC: anti-LYVE-1, anti-VEGFR-3	+
Masuyama et al., 2009 [91]	mice	light microscope, IHC: anti-LYVE-1	+
Gerli et al., 2010 [56]	human	light and electro microscope, Western blotting, method IHC: anti-LYVE-1, anti-VEGFR, D2-40, Prox-1	−
Martin, 2010 [28]	dog	light microscope, IHC: anti-LYVE-1, anti-Prox-1	−
Takahashi et al., 2012 [92]	mice	light microscope, IHC: anti-VEGF-C, anti-VEGF-D, anti-VEGRF-3; anti-vWF	+

Observation: “+” positive result, “−” negative result.

**Table 6 biology-10-01257-t006:** Examples of markers used in immunohistochemical studies.

Antibody	Specificity
VEGF-C	Vascular endothelial growth factor C
VEGF-D	Vascular endothelial growth factor D
VEGFR-2	Vascular endothelial growth factor receptor 2
VEGFR-3	Vascular endothelial growth factor receptor 3
Prox1	Prospero homeobox protein 1
LYVE-l	Lymphatic vessel endothelial hyaluronan receptor 1
D2-40	Podoplanin and O-linked sialoglycoprotein expressed on lymphatic endothelial cells

## Data Availability

No new data were created or analyzed in this study. Data sharing is not applicable to this article.

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
