# Peer review of "Review on the Lymphatic Vessels in the Dental Pulp"

_biology, 2021, doi:10.3390/biology10121257_

Round 1

Reviewer 1 Report

nicely written and thouough overview of the state of the art concerning dental lymphatics

adding a list of perspectives / outstanding research questions at the end of the review would help underline what directions could be envisaged to forward this field relevant to dental decay or loss.

minor comments:

line 164 "arteries and veins are in contact with the lymphatic vessels of adjacent vessels" please clarify the meaning

line 166 "The formation of  lymphatic vessels in response to inflammation is a matter of dispute." perhaps add reference to reviews highlighting that immune cells secret prolymphangiogenic factors and thus stimulation of lymphangiogenesis has been demonstrated under conditions of inflammation in various tissues. (as also stated by the authors on line 199, ref 40)

line 176: " when they are exposed on acids" clarify that the tubules are formed during bacterial acid discharge 

line 186: "tissue-related effects" replace with profibrotic effects

line 188; "circulation of body fluids" replace with:   reuptake of extravasated fluids

line 203: replace "the presence of " with the role of

line 208: "have revealed that inhibition of the VEGF-C/VEGFR-3 axis inhibits lymphangiogenesis" seem superfluous as it is well established fact

line 214: perhaps add that many of these proangiogenic growth factors also may directly or indirectly stimulate lymphangiogenesis

line 252: " Lymphangiogenesis contributes to vascular endothelial growth factors:" change to : Lymphangiogenesis is regulated by vascular endothelial growth factors

line 259: "specific to lymphatic vessels" change to "exclusive to lymphatic vessels"

line 260: "uses specific lymphatic endothelial " change to "uses preferably a combination of lymphatic markers"

line 263: "Lymph cell differentiation" change to "Lymphatic endothelial cell differentiation

line 314: "Podoplanin is administered to podocytes" please clarify

Author Response

Dear Sir or Madam,
We would like to thank you for taking the time and effort necessary to provide such insightful guidance. We carefully considered your suggestions and comments, which further improved the understanding and potential impact of our review. The mistakes were corrected.

Yours faithfully,
Authors

Reviewer 2 Report

Manuscript biology-1469829-peer-review-v1 evaluates the presence of the lymphatic vessels in the dental pulp.

The manuscript is well written and describes an important topic. However, the authors need to consider the following comments:

  1. Page 4, lines 160-163, “Deep carious lesions …….. lymphatic vessels may also appear.” References are required to support these sentences.
  2. Page 4, lines 165-166, “Lymphatic vessels are most frequently formed from already existing ones.” This proposition needs a reference.
  3. Page 4, line 185. TGF-b should read TGF-β.

Author Response

Dear Sir or Madam,
We would like to thank you for taking the time and effort necessary to provide such insightful guidance. We carefully considered your suggestions and comments, which further improved the understanding and potential impact of our review. The references were added. Mistake was corrected.

Yours faithfully,
Authors